# CATS: Contextually-Aware Thresholding for Sparsity in Large Language Models

**Je-Yong Lee**[1]
Mathematical Institute
Oxford University
je-yong.lee@worc.ox.ac.uk

**Donghyun Lee**[1]
Department of Computer Science
University College London
donghyun.lee.21@ucl.ac.uk

**Genghan Zhang, Mo Tiwari, and Azalia Mirhoseini**
Department of Computer Science
Stanford University
{zgh23, motiwari, azalia}@stanford.edu

## Abstract

The dramatic improvements in Large Language Models (LLMs) come at the cost of increased computational resources for inference. Recent studies ameliorate the computational costs of LLMs by increasing their activation sparsity but suffer from significant performance degradation on downstream tasks. In this work, we introduce a new framework for sparsifying the activations of LLMs and reducing inference costs, dubbed Contextually Aware Thresholding for Sparsity (CATS). CATS is a relatively simple algorithm that is easy to implement and highly effective. At the heart of our framework is a new non-linear activation function. We demonstrate that CATS can be applied to various models, including Mistral-7B and Llama2-7B & 13B, and outperforms existing sparsification techniques across multiple tasks. More precisely, CATS-based models achieve downstream task performance within ~99% of their base models at 50% activation sparsity, even without fine-tuning. Moreover, with fine-tuning that targets only 1% of the parameters, CATS-based models not only converge faster but also achieve better task performance than competing techniques. Finally, we develop a custom GPU kernel for efficient implementation of CATS that translates the activation sparsity of CATS to real wall-clock time speedups. Our custom kernel implementation of CATS results in a ~15% improvement in wall-clock inference latency of token generation. We release our code, experiments, and datasets at `https://github.com/ScalingIntelligence/CATS`.

## 1 Introduction

LLMs have demonstrated remarkable success across a variety of fields (Devlin et al., 2018; Brown et al., 2020; Achiam et al., 2023; Brohan et al., 2023), however, this progress comes with significant computational costs. The training of GPT-3 is estimated to have consumed over 3,000,000 GPU-hours and emitted three thousand times the $CO_2$ equivalent of a round-trip flight from San Francisco to New York (Patterson et al., 2021). Furthermore, inference costs often eclipse training costs for models that serve trillions of queries. As such, there is significant interest in reducing the inference costs of LLMs while preserving task performance.

Various techniques have been proposed to mitigate LLM inference costs. These approaches are often based on quantization (Frantar et al., 2022; Dettmers et al., 2022), pruning (Ma et al., 2023; Sun et al., 2023), and other forms of weight sparsification Frantar & Alistarh (2023). Mixture of Experts (MoE) techniques have emerged as particularly promising and

---

[1]These authors contributed equally as co-first authors.

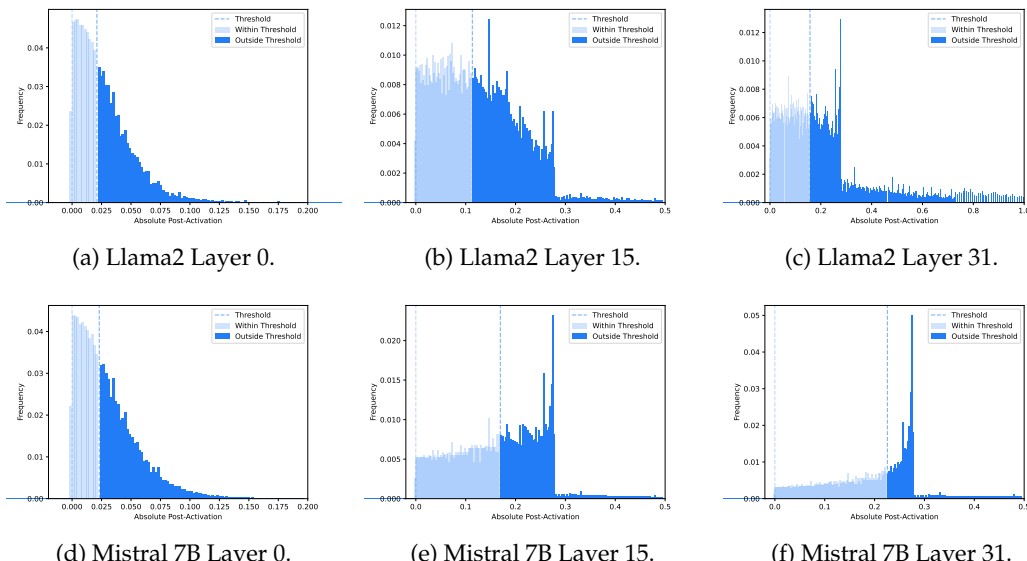

(a) Llama2 Layer 0.      (b) Llama2 Layer 15.      (c) Llama2 Layer 31.

(d) Mistral 7B Layer 0.      (e) Mistral 7B Layer 15.      (f) Mistral 7B Layer 31.

Figure 1: Histograms of post-MLP activations of different layers in different models. Subfigures (a), (b), and (c) correspond to Layers 0, 15, and 31 in Llama2-7B, respectively. Subfigures (d), (e), and (f) correspond to Layers 0, 15, and 31 in Mistral 7B, respectively. The absolute threshold indicates 50% sparsity, where values smaller than the threshold are considered negligible in our technique and thus zeroed out.

are employed by current state-of-the-art LLMs (Shazeer et al., 2017; Lepikhin et al., 2020; Fedus et al., 2022c; Jiang et al., 2024).

MoE techniques activate only a subset of parameters at each inference stage, thereby reducing memory and computational requirements compared to using the entire model. Prevailing implementations of MoE techniques introduce many multi-layer perceptrons (MLPs; the "experts") and dynamically select which experts to multiply with the hidden vector. This selection is performed by a "router"—a small neural network trained to determine the appropriate experts to activate based on the input (Lewis et al., 2021; Rajbhandari et al., 2020).

Concurrently, recent work has observed that activations in the MLP blocks of LLMs are sparse (Liu et al., 2023b; Mirzadeh et al., 2023). This implies that only a few rows (or columns) of the corresponding weight matrices are required for the forward pass. Intuitively, if we could predict *a priori* which elements of the weight matrices were unnecessary via an oracle, we could obviate their respective computations. This is thematically similar to MoE approaches: the activated neurons of the weight matrices can be viewed as activated "experts" and the oracle can be seen as the "router."

We observe that the activation patterns of common LLMs suggest a path to such an oracle. Figure 1 shows a histogram of the post-MLP activations for Layers 0, 15, and 31 for Llama-7B and Mistral-7B on a sample of 500 data points from the RefinedWeb dataset (Penedo et al., 2023). Many of the activations are concentrated about 0; if these activations could be made exactly 0, the corresponding weights of the MLP blocks could be made unnecessary during inference. It is this observation that motivates our study.

In this work, we make the following **contributions**:

1. We draw a connection between the MoE framework and multiplication performed by dense matrices in the MLP blocks of LLMs.

2. We introduce a new sparsification procedure based on a novel activation function, dubbed CATS (for Contextually Aware Thresholding for Sparsity), motivated by an

     empirical evaluation of activation distributions (Figure 1). Crucially, CATS allows for a controllable level of sparsity.

3. We demonstrate that, without any fine-tuning, CATS can be applied to various models, including Mistral-7B and Llama2-7B & 13B, and achieves comparable downstream task performance even at sparsity levels as high as 50%.

4. We demonstrate that, with parameter-efficient fine-tuning, CATS outperforms an existing sparsification technique in downstream task performance at the same sparsity level and number of fine-tuning steps.

5. We provide a custom GPU kernel implementation that exploits the sparsity of CATS and achieves a ~15% improvement in wall-clock inference latency of token generation over the dense models.

## 2 Related Work

Significant recent work focuses on reducing the inference costs of LLMs. Approaches that utilize mixture-of-experts or activation sparsity are most similar to our work.

**Mixture-of-Experts (MoE)** techniques induce effective sparsity in LLMs by determining which subset of subnetworks (the "experts") to activate during the inference pass, often via a trained "router" subnetwork. This is a popular line of work with significant research interest (Shazeer et al., 2017; Hazimeh et al., 2021; Zhou et al., 2022; Lewis et al., 2021; Roller et al., 2021; Zuo et al., 2021; Komatsuzaki et al., 2022; Lou et al., 2021; Mustafa et al., 2022; Rajbhandari et al., 2022; Zhang et al., 2022a;b; Fedus et al., 2022a; Zoph et al., 2022; Kudugunta et al., 2021; Fedus et al., 2022c; Lepikhin et al., 2020; Du et al., 2022; Fedus et al., 2022b; Jiang et al., 2024). For a review of MoE models, we refer the reader to (Fedus et al., 2022a).

**Activation Sparsity:** Activations are known to be sparse in LLMs that utilize ReLU non-linearities in their MLP blocks (Li et al., 2022); however, the reasons for this are not well-understood Hoefler et al. (2021). Nonetheless, activation sparsity induced by ReLU non-linearities has been explored to reduce memory usage and inference time (Jaszczur et al., 2021; Liu et al., 2023b; Szatkowski et al., 2023). Recent work in this area has framed the rows of weight matrices in MLP layers as experts, similar to our work, and/or deploys a small neural network to predict which activations will be non-zero to reduce inference costs (Zhang et al., 2024; Liu et al., 2023b) in these ReLU-based models.

Crucially, however, recent state-of-the-art LLMs such as Mistral-7B (Jiang et al., 2023), Llama2-7B (Touvron et al., 2023), and Gemma (Team et al., 2024)) employ MLP blocks based on more complex nonlinearities that do not inherently induce sparsity Mirzadeh et al. (2023). As such, most of the work on ReLU-based activation sparsity is inapplicable to these models. To the best of our knowledge, ReLUfication is the only work that attempts to bridge this gap (Mirzadeh et al., 2023). ReLUfication replaces the SiLU and GeLU activation functions in LLMs with ReLU to induce sparsity. ReLUfication is the primary baseline against which we compare CATS. In contrast with ReLUfication, CATS contains a controllable level of sparsity. Furthermore, in Section 5, we demonstrate that CATS demonstrates significantly better downstream task performance and fine-tuning efficiency than ReLUfication.

We note that Zhang et al. (2024) is concurrent to our work. In contrast with their work, however, our work is not an empirical evaluation of existing activation functions. Rather, we propose a new framework for sparsifying LLMs. Our framework utilizes a novel activation function and enables controllable sparsity. We validate the performance of CATS in extensive evaluations and provide a custom GPU kernel that translates CATS' sparsity to real wall-clock time gains in Section 5.

We discuss additional research areas on LLM efficiency, such as quantization, structure pruning, knowledge distillation, and hardware-aware optimization in Appendix A.

# 3   Background

**Motivation:** As described in Section 1, MoE models selectively activate expert subnetworks via a trained router. Crucially, we may view the rows (or columns) of MLP matrices as experts in an MoE model. To identify the layers most likely to benefit from this MoE perspective (where many activations can be zeroed), we examine the activations of different layers in LLMs. Figure 1 demonstrates that activations of the Gated-MLP layers tend to concentrate around zero across different LLMs. This behavior suggests that many neurons of MLP layers minimally affect the output in future operations.

**Gated-MLP Blocks:** We now describe the components of LLMs that our work aims to accelerate: the Gated-MLP blocks. They are commonly used in LLMs, including in the Llama2 family of models, Mistral-7B, and Gemma. A Gated-MLP block consists of several fully-connected layers and performs the following computation:

$$\text{Gated-MLP}(x) \coloneqq (\text{SiLU}(xW_{\text{gate}}) * (xW_{\text{up}}))W_{\text{down}} \tag{1}$$

where $x \in \mathbb{R}^{b \times d}$, $W_{\text{gate}}, W_{\text{up}} \in \mathbb{R}^{d \times m}$, $W_{\text{down}} \in \mathbb{R}^{m \times d}$, $*$ indicates element-wise multiplication, and

$$\text{SiLU}(x) \coloneqq x * \text{sigmoid}(x) = \frac{x}{1 + e^{-x}} \tag{2}$$

Crucially, the operation $\text{SiLU}(xW_{\text{gate}})$ can be viewed as the router in an MoE model. Through this lens, the columns of $W_{\text{up}}$ and the rows of $W_{\text{down}}$ are the experts. If $\text{SiLU}(x)$ is always binary, i.e., 1 or 0, it would turn on/off elements of the remaining computation (multiplication by $W_{\text{up}}W_{\text{down}}$). When $\text{SiLU}(x)$ is not binary, it can be viewed as a "soft" router that weighs the experts by different amounts.

# 4   Method: Contextually-Aware Thresholding for Sparsification (CATS)

We now describe CATS, a framework to accelerate the Gated-MLP blocks of LLMs. The CATS framework proposes a new, simple activation function and exploits the sparsity induced by this activation. In Section 5, we apply CATS to Mistral-7B and Llama2-7B and show that CATS-based models still exhibit significant activation sparsity, even when fine-tuned.

## 4.1   Stage 1: Determining Cutoff Threshold

We assume we are given a desired sparsity level $k$ (e.g., 70%) as input. For each Gated-MLP block in the LLM, we compute the activations over a random subset of the training data, limited to only 500 data points. We then compute the *cutoff threshold* as the $k$th percentile of the resulting values.

Concretely, the cutoff threshold $t$ is

$$t \coloneqq \min\{t' : F(t') \geq k\} \tag{3}$$

where $F$ represents the empirical cumulative distribution function of the absolute values of the activations for the given MLP block.

Figure 1 shows histograms of the absolute values of activations for different MLP blocks in various models on the RefinedWeb dataset (Penedo et al., 2023). A sparsity level of 70% corresponds to a threshold of approximately 0.15; different sparsity levels correspond to different thresholds. We note that these thresholds are chosen and fixed before any further fine-tuning.

## 4.2 Stage 2: Sparsifying Gate-MLP Blocks

Given the cutoff threshold $t \geq 0$ corresponding to the input sparsity level $k$, we wrap the SiLU($x$) activations in each MLP block with the CATS activation. The CATS operation, denoted as $\text{CATS}_t(\cdot)$, is defined as:

$$\text{CATS}_t(\mathbf{x_j}) := \begin{cases} x_j, & \text{if } |x_j| \geq t \\ 0, & \text{if } |x_j| < t \end{cases} \tag{4}$$

Here, $t$ is the sparsification threshold and $x_j$ is the $j$-th element of the vector $\mathbf{x}$, respectively.

This results in a new activation $\text{CATS}_t(\text{SiLU}(\cdot))$:

$$\text{CATS}_t(\text{SiLU}(xW_{\text{gate}})) = \begin{cases} \text{SiLU}(xW_{\text{gate}}) & |\text{SiLU}(xW_{\text{gate}})| \geq t \\ 0 & |\text{SiLU}(xW_{\text{gate}})| < t \end{cases} \tag{5}$$

Intuitively, the resulting model zeros out activations that are likely to be close to 0 because their corresponding inputs were small. This procedure produces a trained model with sparse activations, whose performance can then be evaluated. We empirically validate that this procedure results in a model with an approximate sparsity level of $k$, even after fine-tuning, as detailed in Appendix C.

## 4.3 Custom Kernel Design

The previous subsections describe the procedure for sparsifying LLM's activations, obviating unnecessary computations, and reducing the required number of floating point operations (FLOPs) in each MLP block. We now translate this reduction in FLOPs to a reduction in actual wall-clock latency and increase in generation throughput via a custom GPU kernel.

We focus on reducing the latency of the MLP blocks by reducing memory accesses, as the MLP blocks are known to be memory-bound during inference when the batch size is small (Kim et al., 2023). As shown in Line 4 of the Custom GPU Kernel 1, we first fuse the element-wise multiplication of $v[\texttt{Mask}]$ into each tiling of $xW_{\text{up}}[\texttt{Mask}]$. Here, $v$ represents the hidden vector after applying the SiLU activation function, and $\texttt{Mask}$ denotes a binary

---

**Custom GPU Kernel 1** MLP using CATS

1: **Input:** threshold $t > 0$, hidden layer $x$, weights $W_{\text{gate}}$, $W_{\text{down}}$, and $W_{\text{up}}$
2: $v \leftarrow \text{SiLU}(xW_{gate})$
3: $\texttt{Mask} \leftarrow 1$ if $|v| \geq t$ else 0
4: $x_1 \leftarrow (xW_{\text{up}}[\texttt{Mask}] * v[\texttt{Mask}])$
5: $y \leftarrow x_1 W_{\text{down}}[\texttt{Mask}]$

---

mask identifying elements of $v$ with large absolute values. This fusion saves memory operations that would be necessary for storing and loading $x_1$ several times. We then directly use $\texttt{Mask}$ to control which parts of the weight matrices $W_{\text{up}}$ and $W_{\text{down}}$ to load, instead of using the compressed indices directly as in Zhang et al. (2023) This further improves the kernel speed because it avoids expensive synchronization operations. In Section 5.3, we demonstrate how our custom GPU kernel effectively reduces the inference latency of CATS-based models as sparsity increases.

# 5 Experiments

In this section, we describe the experiments with which we assess the performance of CATS. We first describe the experimental details that are common to all experimental settings. We then describe experiments on downstream task performance. Finally, we measure CATS' effect on wall-clock time inference when implemented with the custom GPU kernel from Section 4. We find that CATS-based models outperform their ReLUfication versions in downstream task performance, with or without fine-tuning, and can exploit their sparsity for wall-clock inference time speedups over the base models.

We first describe the experimental setup, including base models, CATS-based models, metrics, datasets, and computational environment.

**Base Models:** We apply CATS to both Mistral-7B and Llama2-7B as base models to verify it is generally applicable to different LLMs. We measure the performance of each CATS model against the original base model. We also compare the performance to of the CATS-based models to the base model transformed by ReLUfication from Mirzadeh et al. (2023).

**CATS-based Models:** For a given base model, we train three CATS-based variants that exhibit different sparsity levels in the MLP blocks: 50%, 70%, and 90% activation sparsity. We call these models CATS-50%, CATS-70%, and CATS-90%, respectively, where the base models are clear from context.

**Metrics:** We compare models using several metrics. In the first set of experiments, we compare each model's accuracy on downstream tasks. In the second set of experiments, we compare each model's wall-clock time inference latency.

**Datasets:** For the downstream task performance experiments, we use the OpenBookQA, ARC_Easy, Winogrande, HellaSwag, ARC_Challenge, PIQA, BoolQ, and SCI-Q datasets from the Eleuther AI Evaluation Harness (Gao et al., 2023) as in Mirzadeh et al. (2023) for ease of comparison; these tasks were originally chosen to measure various abilities of the models across various domains, such as reading comprehension and reasoning. For the latency and generation task experiments, we assess the wall-clock inference time and perplexity score on the RefinedWeb test dataset (Penedo et al., 2023).

**Computational Environment:** All experiments were run on a single machine with 8 L40S GPUs. Latency experiments were run on a single L40S GPU as each 7B base model was able to fit in a single GPU RAM when performing inference in brain float 16 (BF16) or floating point 16/32 (FP16/32) precision. We used DeepSpeed (Rasley et al., 2020) with BF16 precision to manage the high memory overhead during training. We also employed Low-Rank Adaptation (LoRA) (Hu et al., 2021) and targeted 1% of the parameters (Query and Key in attention modules, $W_{gate}$, and $W_{down}$) in the fine-tuning experiments. During inference, we used the `transformers v4.36.2` HuggingFace library, `PyTorch v2.1.2`, and `CUDA v12.1`. We used `Triton v2.1.0` for our GPU kernels. All experiments were run in FP32 precision; changing this to FP16 did not materially affect results. All of our code, including a one-line script to set up an environment and reproduce all of our results, is available in the supplementary material.

## 5.1 Downstream Task Performance

We now compare the downstream task performance of the CATS-based models to the baseline models in several settings and draw several conclusions.

**CATS-based models perform comparably to the base models and outperform ReLUfication in zero-shot accuracy without any fine-tuning:** We first compare the performance of CATS-based models to the baseline models without any fine-tuning. In this setting, the CATS prescription is applied directly to the base models, i.e., the activation functions are simply replaced in the MLP blocks and no fine-tuning is performed. Table 1 shows our results across 8 different benchmark tasks. CATS-based models demonstrate performance comparable to the unchanged, out-of-the-box base models, even at high sparsity levels. In particular, at CATS 50% demonstrates performance comparable to the base model. CATS significantly outperforms ReLUfication in downstream task performance at the same sparsity level (90%).

**CATS-based models perform more comparably to the base model and increasingly outperform ReLUfication as the model size increases in zero-shot accuracy:** Similar to the previous evaluation, we assess the performance of various CATS configurations and a baseline Llama-13B model across eight downstream tasks without any fine-tuning. Notably, the performance degradation of CATS-50% decreased from 1.46% at 7B to 0.65% at 13B. Conversely, the performance gap between CATS-50% and ReLUfication increased from 29.08% at 7B to 34.55% at 13B. We present the detailed results in Appendix E.

| Model \ Dataset | WG acc↑ | PIQA acc↑ | SciQ acc↑ | QA acc↑ | HS acc↑ | BoolQ acc↑ | Arc-E acc↑ | Arc-C acc↑ | Avg acc↑ |
|---|---|---|---|---|---|---|---|---|---|
| **Mistral-7B** | 0.7419 | 0.8069 | 0.959 | 0.3260 | 0.6128 | 0.8370 | 0.8085 | 0.5034 | 0.6994 |
| CATS 50% | 0.7245 | 0.8009 | 0.948 | 0.3200 | 0.6097 | 0.8193 | 0.7849 | 0.5043 | 0.6890 |
| CATS 70% | 0.7190 | 0.8003 | 0.929 | 0.292 | 0.6057 | 0.8028 | 0.7492 | 0.4693 | 0.6709 |
| CATS 90% | 0.5627 | 0.6001 | 0.422 | 0.212 | 0.3359 | 0.7086 | 0.3754 | 0.2773 | 0.4368 |
| ReLUfication | 0.5043 | 0.5092 | 0.236 | 0.142 | 0.2580 | 0.4208 | 0.2723 | 0.2415 | 0.3230 |
| **Llama2-7B** | 0.6906 | 0.7807 | 0.94 | 0.314 | 0.5715 | 0.7774 | 0.7630 | 0.4343 | 0.6589 |
| CATS 50% | 0.6748 | 0.7693 | 0.927 | 0.322 | 0.5711 | 0.7263 | 0.7441 | 0.4121 | 0.6433 |
| CATS 70% | 0.6693 | 0.7584 | 0.902 | 0.294 | 0.5500 | 0.6590 | 0.7008 | 0.3805 | 0.6143 |
| CATS 90% | 0.5738 | 0.6627 | 0.611 | 0.212 | 0.3848 | 0.6284 | 0.4566 | 0.2816 | 0.4764 |
| ReLUfication | 0.4893 | 0.5408 | 0.2570 | 0.154 | 0.2586 | 0.6003 | 0.2795 | 0.2406 | 0.3525 |

Table 1: Downstream task performance of base models, CATS-based models at varying levels of sparsity, and ReLUfication across different benchmark datasets. The CATS versions of the base models demonstrate comparable performance to original models at high sparsity levels, e.g., 50%. At higher sparsity levels, CATS still outperforms ReLUfication.

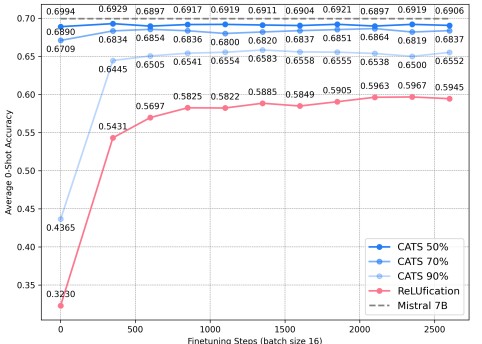 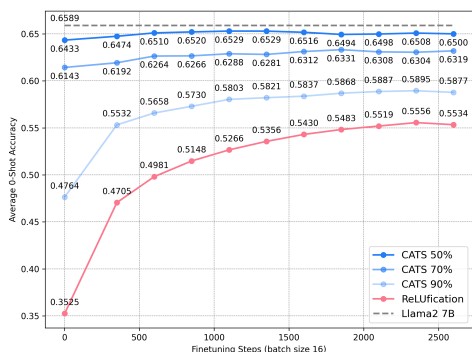

Figure 2: Downstream task performance of the base model, CATS models with different sparsity levels, and ReLUfication across varying numbers of fine-tuning steps on the RefinedWeb dataset applied to Mistral-7B (left) and Llama2-7B (right). The CATS models exhibit faster convergence and greater fine-tuning efficiency than the ReLUfication variants. Furthermore, CATS-50% and CATS-70% demonstrate comparable performance to the base models without any fine-tuning (0 fine-tuning steps).

**CATS-based models perform comparably to the base models and outperform ReLUfication in zero-shot accuracy with "general" fine-tuning:** In this setting, CATS is applied the base models Llama-7B and Mistral-7B. All models are then fine-tuned using LoRA (Hu et al., 2021), targeting only 1% of the parameters, on the RefinedWeb dataset Penedo et al. (2023). Their downstream performance is subsequently measured across 8 evaluation datasets. We emphasize that the dataset upon which the models are fine-tuned is different from the evaluation datasets in this setting. Figure 2 demonstrates our results. We note several key observations:

1. CATS-based models still exhibit sparsity after fine-tuning (see Appendix C).

2. CATS-50% demonstrates performance comparable to the base models, even without any fine-tuning. This is in contrast with ReLUficiation, which demonstrates poor performance without fine-tuning.

3. CATS-50%, CATS-70%, and CATS-90% all display better task performance than ReLUfication when controlling for the number of fine-tuning steps. In particular, even with very few fine-tuning steps, the CATS-based models achieve comparable performance to the base models.

4. CATS-based models, even with sparsity levels as high as 70%, achieve performance comparable to the base models within 500 steps of fine-tuning, whereas ReLUfication does not.

**CATS-based models perform comparably to the base models and outperform ReLUfication with task-specific fine-tuning but without "general" fine-tuning:** In this setting, the CATS prescription is applied to Mistral-7B. All variants are then fine-tuned for 10 epochs on the training data and evaluated on test dataset for the Cola, SST2, and BoolQ datasets. Table 2 demonstrates our results. Our observations are similar to those for "general" fine-tuning:

1. CATS-based models still exhibit sparsity after fine-tuning (see Appendix C).

2. CATS-50% demonstrates performance comparable to the base models. This is in contrast with ReLUficiation, which demonstrates a significant performance degradation.

3. CATS-50%, CATS-70%, and CATS-90% all display better task performance than ReLUfication.

| Dataset/Sparsity | Base Model | 0.5 | 0.7 | 0.9 | ReLUfication |
|:---:|:---:|:---:|:---:|:---:|:---:|
| Cola | **0.8667** | 0.8658 (-0.10%) | 0.8552 (-1.32%) | 0.8303 (-4.21%) | 0.6922 (-20.13%) |
| SST2 | 0.9644 | 0.9656 (+0.12%) | **0.9702** (+0.60%) | 0.9427 (-2.25%) | 0.7856 (-18.55%) |
| BoolQ | **0.8905** | 0.8862 (-0.48%) | 0.8807 (-1.10%) | 0.7920 (-11.06%) | 0.6624 (-25.61%) |
| Average | **0.9072** | 0.9059 (-0.13%) | 0.9020 (-0.52%) | 0.8550 (-5.22%) | 0.7134 (-19.38%) |

Table 2: Downstream task performance of Mistral-7B and its CATS-based and ReLUfication variants across three different benchmark datasets. Top accuracies are marked in bold and second-highest in underline. Relative performance degradation is given in parentheses. CATS-50% demonstrates performance within 0.5% of the base model, whereas ReLUfication demonstrates a significant performance drop.

## 5.2 Generation task performance

We now evaluate the generation task performance of the CATS-based models using the RefinedWeb dataset. Without any fine-tuning, CATS-50% exhibits only a slight increase in perplexity scores. The difference in perplexity scores between the base models and CATS-50% further diminishes after 1300 fine-tuning steps.

Specifically, for Llama-7B, the perplexity increases by just 1.06% when comparing the base model to CATS-50% without fine-tuning, and this difference narrows to 0.60% with minimal fine-tuning. Similarly, for Mistral-7B, the initial perplexity increase of 1.21% is reduced to a mere 0.45% after fine-tuning.

| Model | Base Model | CATS 50% | CATS 50%* | ReLUfication | ReLUfication* |
|:---:|:---:|:---:|:---:|:---:|:---:|
| Llama-7B | **2.18** | 2.203 (+1.06%) | 2.193 (+0.60%) | 3.104 (+42.38%) | 2.617 (+20.04%) |
| Mistral-7B | **2.234** | 2.261 (+1.21%) | 2.244 (+0.45%) | 2.801 (+25.34%) | 2.562 (+14.68%) |

Table 3: Perplexity scores for different models and configurations. The best score is marked in bold and the second-best is underlined. Percentage differences relative to the base model are shown in parentheses. Models marked with * are fine-tuned for 1300 steps.

## 5.3 Wall-clock Time Speedups for Inference

Activation sparsity in a model is not sufficient to directly achieve wall-clock time inference speedups (Frantar & Alistarh, 2023). In this subsection, we demonstrate that our custom GPU kernel translates the activation sparsity induced by CATS into tangible wall-clock time improvements.

**CATS-based models can translate their activation sparsity to wall-clock time speedups:**

Figure 3 shows the wall-clock inference time of the dense model compared to CATS implemented via the custom GPU kernel described in Section 4.3, for various sparsity levels of

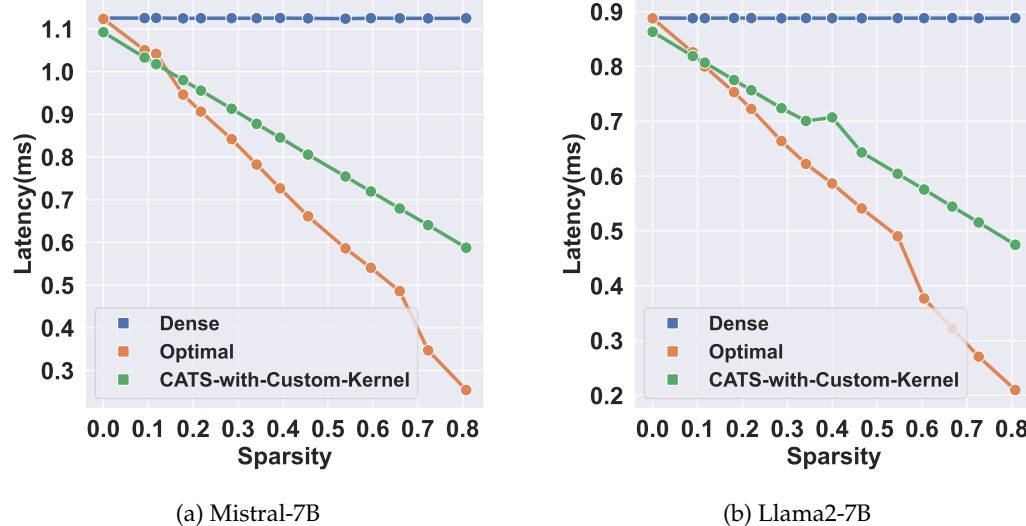

(a) Mistral-7B

(b) Llama2-7B

Figure 3: Latency of the original Mistral-7B MLP block (left, "Dense"), Llama-7B MLP block (right, "Dense"), and their CATS-based variants at different sparsity levels, compared to "Optimal." Our custom GPU kernel improves the latency of the CATS-based variants and achieves performance close to "Optimal" for most reasonable sparsity levels.

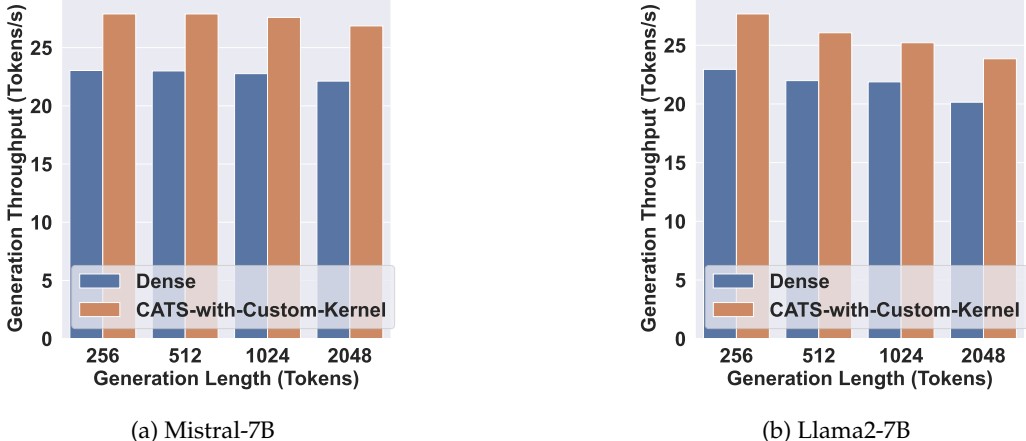

(a) Mistral-7B

(b) Llama2-7B

Figure 4: Throughput of Mistral-7B (left, "Dense") and Llama2-7B (right, "Dense") and CATS-50% with the custom GPU kernel. CATS-50% demonstrates significantly higher throughput.

CATS. We evaluate the latency of a single MLP block and the throughput of the generation stage of the end-to-end inference. Mistral-7B contains 32 MLPs with $m = 14336$ and $d = 4096$, and Llama2-7B contains 32 MLPs with $m = 11008$ and $d = 4096$ ($m$ and $d$ are defined after Equation 1).

In Figures 3a and 3b, we compare our method ("CATS-with-Custom-Kernel") with the dense MLP with $m_{dense} = m$ ("Dense") and the dense MLP with $m_{optimal} = m * Sparsity$ ("Optimal"), the latter of which is a proxy for the best wall-clock time we could hope to achieve. At 50% and 70% sparsity, the sparse kernel achieves approximately a 40% and 70% speedup, respectively, over the original dense MLP. Latency measurements were obtained by conducting 20 rounds of warmups, repeating the kernel 80 times, and computing the geometric mean of the latency across each round. The comparison with Dense demonstrates that our sparse kernel consistently outperforms the original MLP. The comparison with

Optimal shows that our sparse kernel closely approaches the Optimal performance at low sparsity levels. However, as the sparsity level increases, the gap between our performance and Optimal widens, as expected. We note that our sparse kernel performs the same number of memory accesses as the Optimal; however, due to differences in access patterns, the methods result in different wall-clock time measurements. Optimal can perform worse than our sparse kernel when $m_{optimal}$ does not match the shapes optimized by GPU libraries (Tillet & Cox, 2017). Conversely, our sparse kernel can underperform compared to Optimal when the overhead of operations on zero values outweighs the benefits of reduced memory access.

In Figures 4a and 4b, we compare dense models with CATS-with-Custom-Kernel (50% sparsity) on the throughput of the generation stage. The generation stage (or "decoding" stage) is known to be memory-bound (Kim et al., 2023), which suggests CATS can improve inference througput. We test the generation throughput at a batch size of 1 and beam width of 1, and record the latency from the first generated token to the last token. The throughput is calculated by the generated length divided by latency. The final throughput is averaged (geometric mean) over 50 samples from the RefinedWeb test dataset. CATS can accelerate the generation stage by ∼18% for Llama2-7B and ∼21% for Mistral-7B at 50% sparsity.

Though we only test on Huggingface (Wolf et al., 2020), our methodology is orthogonal to the framework and thus can be used in other LLM serving systems such as DeepSpeed (Rajbhandari et al., 2022) and TensorRT-LLM (Nvidia, 2024).

## 6 Discussion and Conclusion

We presented CATS, a novel framework for inducing and exploiting activation sparsity in LLMs. At the heart of our framework is the CATS activation, given in Equation 5, that induces a controllable level of activation sparsity in LLMs. We also provide a custom GPU kernel implementation that exploits CATS's sparsity to achieve real wall-clock time gains in inference latency.

CATS-based models demonstrate downstream task performance comparable to unmodified base models and better than baseline models with no fine-tuning, even at sparsity levels as high as 50%. CATS-based models also exhibit better behavior than ReLUfication at similar levels of fine-tuning, and often achieve performance comparable to the base model at high levels of sparsity, both with general and task-specific fine-tuning.

**Limitations and Future Work:** Our work leaves several opportunities for future work. Most importantly, our empirical evaluations of CATS were restricted to the Mistral-7B and Llama2-7B base models. While we suspect CATS would also apply to other, larger models, we leave a precise empirical study to future studies. Future work may also investigate how to apply techniques similar to CATS to other MLP architectures beyond Gated-MLP, or to attention layers but without a task performance degradation. It may be possible, for example, to use recent techniques to accelerate attention layers (such as those from Zhang et al. (2022a) and Voita et al. (2019)) in conjunction with CATS.

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

## A    Additional Related Work

In this appendix, we discuss additional veins of related work.

**Hardware-Aware Optimization** that relies on customizing the algorithm implementation for the underlying hardware can result in significant performance speedup Dao et al. (2022); Fu et al. (2023a), especially for sparse kernels Gale et al. (2023); Yu et al. (2023). Recent hardware-aware methods in LLMs have shown to be highly effective in lowering the cost of attention operation Rabe & Staats (2022); Dao (2023); Liu et al. (2023a). Similar to attention operation, MLP is also memory-bounded on highly parallel machines like GPU Kim et al. (2023). The sparsity has the potential to expedite MLP because it can increase the arithmetic intensity. Based on the Roofline analysis Williams et al. (2009), higher arithmetic intensity means shorter wall-clock time for memory-bounded operations. In this work, we focus on leveraging the sparsity to reduce the memory transfers associated with the MLP weights. We do so by designing algorithmic optimizations that adaptively induce sparsity and implementing hardware-aware optimizations that translate the achieved nominal sparsity into actual wall-clock time speedup.

**Structural pruning** techniques induce sparsity by setting certain weights to zero so their corresponding activations need not be computed Wang et al. (2019); Kurtic et al. (2022); Xia et al. (2022); Zafrir et al. (2021); Ma et al. (2023). However, applying such techniques naïvely may not result in actual wall-clock time speedups if the resulting sparsity pattern does not lower the number of General Matrix Multiplication (GEMM) calls. Furthermore, the pruning pattern is determined at the model level and is not adaptive to the inputs, which may result in a degradation in task performance.

**Quantization and Knowledge Distillation** from larger models to smaller models are other popular forms of LLM inference optimization Bai et al. (2020); Frantar et al. (2022); Dettmers et al. (2023); Sun et al. (2019; 2020); Pan et al. (2020); West et al. (2022); Fu et al. (2023b). These methods often reduce the memory and computational complexity at the cost of performance degradation or require extensive finetuning. Our work can be applied to quantized or distilled models as well, although the achieved sparsity level on these models may differ.

# B   Accelerating Attention Layers

## B.1   Method

In this section, we discuss how we can apply CATS to reduce the inference costs of attention layers inside Transformer blocks. The basic operations of a Transformer block can be written as:

$$\text{MLP}_i(\text{Attention}_i(\mathbf{x})) \tag{6}$$

where $\mathbf{x}$ is the hidden vector right before the $i$-th layer and where we have excluded operations like batch normalization, positional embedding, residual connections, etc. for simplicity. (For more details on the variants of Attention layers and those used in our models, we refer the reader to Touvron et al. (2023) and Jiang et al. (2023).)

The new equation for $i$-th transformer layer, where we wrap the previous layer with CATS activations, becomes:

$$\text{MLP}_i(\text{CATS}_{t_{i,1}}(\text{Attention}_i(\text{CATS}_{t_{i,2}}(\mathbf{x})))) \tag{7}$$

where $t_{i,1}$ and $t_{i,2}$ are the sparsification thresholds for the CATS operations applied before the MLP and attention layers, respectively, in the $i$-th transformer layer.

We verify that this operation results in sparse activations in Appendix C.

## B.2   Experimental Results

**CATS can also be applied to accelerate the attention blocks of LLMs**: We also apply CATS to accelerate the computation of attention layers. Our approach is inspired by "Stage 2" of ReLUfication (Mirzadeh et al., 2023).

Due to space constraints, we only measure the performance of CATS-50% applied to the base Mistral-7B model and measure zero-shot task performance. We fine-tune both models for 2000 fine-tuning batches of 16 examples each. Stage 2 CATS, which appplies CATS to both the MLP and Attention blocks, demonstrates an average downstream task performance of 66.84% across the 8 different evaluation tasks from Section 5, whereas the base Mistral-7B model demonstrates an average task performance of 69.94%. In contrast, the original CATS, applied only to the MLP layers, demonstrates an average task performance of 69.21%.

Our results demonstrate that CATS can also be applied to the attention layers of LLMs, albeit with a slight (4.3% relative) performance degradation. Future work may investigate how to apply CATS in way that better preserves the performance of the model.

# C   Target sparsity vs. actual sparsity

Figure 5 demonstrates the the sparsity of each layer of Mistral-7B and Llama2-7B after CATS has been applied and fine-tuning has been performed on the RefinedWeb dataset. The average sparsity of each model (dashed lines) is roughly equal to the target sparsities (indicated by the legend).

Table 4 demonstrates the average layer sparsity of each model after task-specific fine-tuning on the 3 datasets used for this experimental setting in Section 5. The observed sparsity levels are approximately equal to the target sparsity levels.

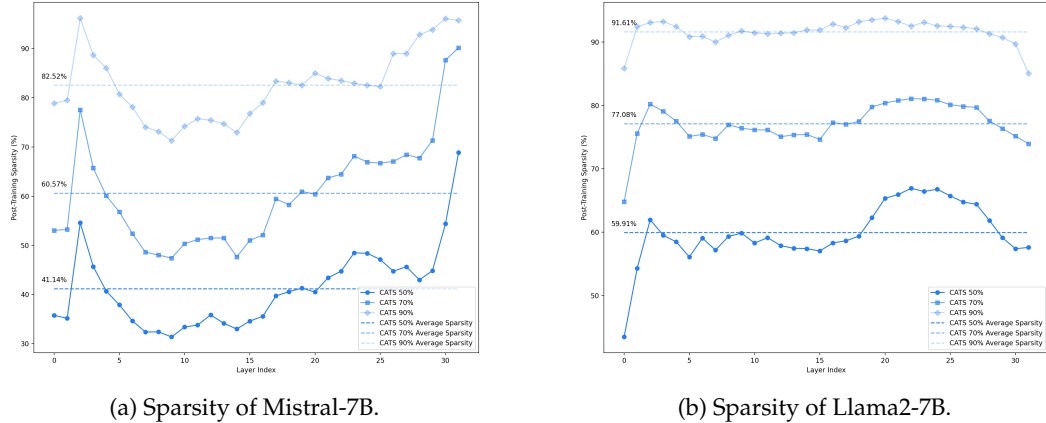

(a) Sparsity of Mistral-7B.

(b) Sparsity of Llama2-7B.

Figure 5: CATS-based models still exhibit sparsity after general fine-tuning on the Refined-Web dataset.

| Dataset/Sparsity | 0.5 | 0.7 | 0.9 |
|---|---|---|---|
| Cola | 49.629 | 68.926 | 87.6 |
| BoolQ | 49.196 | 68.444 | 87.571 |
| SST2 | 48.727 | 68.738 | 87.882 |
| Average | 49.184 | 68.703 | 87.684 |

Table 4: CATS-based models' final sparsity after specific fine-tuning on each task. They continue to exhibit sparsity after task-specific fine-tuning.

Future work might focus on enforcing a minimum sparsity layer-wise, i.e., by zeroing out at least enough neurons to enforce the desired sparsity level $k$ for each layer. Such work could investigate the tradeoffs between sparsity, latency, and downstream task performance.

## D   Details on Custom GPU Kernel Design

The previous subsections describe the procedure by which we sparsify the activations of an LLM, obviate some computations, and reduce the required number of FLOPs. Though significant recent work has focused on FLOPs as a proxy for inference cost, other work has demonstrated that reducing FLOPs is not sufficient to reduce real wall-clock inference latency Liu et al. (2023b). However, predictable sparsity patterns can be exploited to reduce floating point operations (FLOPs) during inference. We now translate the reduction in FLOPs to an actual wall-clock latency reduction via several custom GPU kernel optimization techniques.

The operations of the Gated-MLP with the CATS activation functions are:

$$v = \text{CATS}(\text{SiLU}(xW_{gate})) \tag{8}$$

$$\text{Mask} = \mathbb{1}_{\{|v|>t\}} \ \text{(elementwise)} \tag{9}$$

$$y = (v' * (xW'_{\text{up}}))W'_{\text{down}} \tag{10}$$

where $v'$, $W'_{\text{up}}$, and $W'_{\text{down}}$ are $v$, $W_{\text{up}}$, and $W_{\text{down}}$ masked by Mask (for the matrices $W_{\text{up}}$ and $W_{\text{down}}$, the entire column $j$ is 0 if $\text{Mask}_j = 0$, i.e., the mask is broadcast across columns).

If Mask is sparse, then Equation (10) performs two sparse matrix multiplications. In fact, only coordinates (respectively, rows) of $v$ (respectively, $W_{\text{up}}$ and $W_{\text{down}}$) corresponding to nonzero coordinates of Mask need to be loaded into memory. Since the MLP layer at inference time is known to be memory-bound Kim et al. (2023), the latency can be reduced

if the memory access is reduced. We exploit these observations to translate the reduction in FLOPs to a real wall-clock time reduction in inference.

---

**Custom GPU Kernel 2** MLP using CATS

1: **Input:** threshold $t > 0$, hidden layer $x$, weights $W_{\text{gate}}$, $W_{\text{down}}$, and $W_{\text{up}}$
2: $v \leftarrow \text{CATS}(\text{SiLU}(xW_{gate}))$
3: $\texttt{Mask} \leftarrow 1$ if $|v| \geq t$ else $0$
4: $\texttt{idcs} \leftarrow$ indices where $\texttt{Mask} = 1$
5: $x_1 \leftarrow (xW_{\text{up}}[\texttt{idcs}] * v[\texttt{idcs}])$
6: $y \leftarrow x_1 W_{\text{down}}[\texttt{idcs}]$

---

**Custom GPU Kernel 3** MLP using CATS without atomic operations

1: **Input:** threshold $t > 0$, hidden layer $x$, weights $W_{\text{gate}}$, $W_{\text{down}}$, and $W_{\text{up}}$
2: $v \leftarrow \text{CATS}(\text{SiLU}(xW_{gate}))$
3: $\texttt{Mask} \leftarrow 1$ if $|v| \geq t$ else $0$
4: $x_1 \leftarrow (xW_{\text{up}}[\texttt{Mask}] * v[\texttt{Mask}])$
5: $y \leftarrow x_1 W_{\text{down}}[\texttt{Mask}]$

---

Algorithms 2 and 3 describe Equations (8)-(10) in lower-level pseudocode. Algorithms 2 and 3 contain several optimizations.

**Optimization 1**: We fuse the element-wise multiplication of $v[\texttt{idcs}]$ into each tiling of $xW_{\text{up}}[\texttt{idcs}]$ as shown in Line 5 of Algorithm 2. We use an efficient algorithm from Deja Vu Liu et al. (2023b) to compute $x_1 = xW_{\text{up}}[\texttt{idcs}]$ without the element-wise multiplication by $v[\texttt{idcs}]$. In this manner, we fuse several operations and save the memory operations for storing and loading $x_1$ several times.

The atomic operations in Line 4 of 2, however, introduce extra overhead. Line 4 compresses a one-hot mask to a compressed coordinate array and requires atomically appending to the $\texttt{idcs}$. GPUs, however, cannot efficiently perform such atomic operations because of their massively parallel nature.

**Optimization 2**: We therefore introduce another optimization in Algorithm 3 to reduce the memory loading incurred by the atomic operations. In Algorithm 3, we directly use $\texttt{Mask}$ to control which parts of weight matrices to load, instead of the condensed $\texttt{idcs}$. Algorithm 3 has more operations than Algorithm 2 because it directly assigns the unloaded elements to zero instead of squeezing out the zero values before computation. Algorithm 3 does not skip the zero operations in a fine-grained way because the sparsity in this problem is not asymptotically high Zhang et al. (2023), which means the operation reduction does not compensate for the performance loss caused by complex control logic. Figure 6 the ablation experiment results

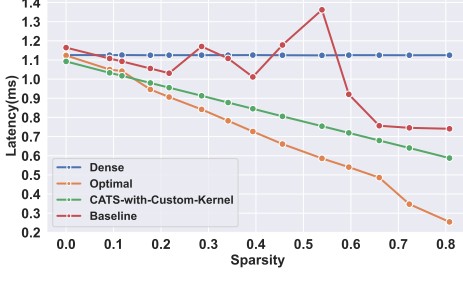 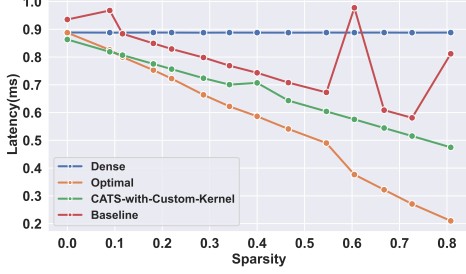

(a) CATS of Mistral-7B's MLP.  (b) CATS of Llama2-7B's MLP.

Figure 6: Ablation study on kernel optimizations.

```
 1  # PyTorch                                              34      X += BLOCK_N
 2  import triton.language as tl                           35  V = V + rm
 3  V = SiLU(X @ W_gate)                                   36  v = tl.load(V, mask=flag, other=0.0)
 4  Mask = torch.abs(V) >= threshold                       37  acc = acc * v.to(tl.float32)
 5                                                         38  X_1 = X_1 + rm
 6  # Triton 1: x_1 = (x @ W_up[Mask]) * v[Mask]           39  tl.store(X_1, acc.to(tl.float16), mask=rm < M)
 7  # Input                                                40
 8  X: [N]                                                 41  # Triton 2: y = x_1 @ W_down[Mask]
 9  V: [M]                                                 42  # Input
10  W_up: [M,N] # Stored in column-major                   43  X_1: [M]
11  Mask: [M]                                              44  W_down: [M,N]
12  # Output                                               45  Mask: [M]
13  X_1: [M]                                               46  # Output
14  # Tunable parameters                                   47  Y: [N]
15  BLOCK_M = {4,8,16,32}                                  48  # Tunable parameters
16  BLOCK_N = {256,512,1024}                               49  BLOCK_M = {16,32,64,128}
17  # Configuration                                        50  BLOCK_N = {128,256,512,1024}
18  grid = (M // BLOCK_M,)                                 51  # Configuration
19  # Kernel[grid]                                         52  grid = (M // BLOCK_M, N // BLOCK_N)
20  start_m = tl.program_id(0)                             53  # Kernel[grid]
21  rm = start_m * BLOCK_M + tl.arange(0, BLOCK_M)         54  start_m = tl.program_id(0)
22  rn = tl.arange(0, BLOCK_N)                             55  start_n = tl.program_id(1)
23  Mask = Mask + rm                                       56  rm = start_m * BLOCK_M + tl.arange(0, BLOCK_M)
24  flag = tl.load(Mask) > 0                               57  rn = start_n * BLOCK_N + tl.arange(0, BLOCK_N)
25  W_up = W_up + (rm[:,None] * N + rn[None,:])            58  Mask = Mask + rm
26  X = X + rn                                             59  flag = tl.load(Mask) > 0
27  acc = tl.zeros((BLOCK_M,), dtype=tl.float32)           60  W_down = W_down + (rm[:,None] * N + rn[None,:])
28  i_mask = flag[:,None]                                  61  X_1 = X_1 + rm
29  for _ in range(N, 0, -BLOCK_N):                        62  w = tl.load(W_down, mask=flag[:,None], other=0.0)
30      w = tl.load(W_up, mask=i_mask, other=0.0)          63  x = tl.load(X_1)
31      x = tl.load(X)                                     64  acc = tl.sum(a * x[:,None], 0).to(tl.float32)
32      acc += tl.sum(w * x[None,:], 1).to(tl.float32)     65  Y = Y + rn
33      W_up += BLOCK_N                                    66  tl.atomic_add(Y, acc.to(tl.float16))
```

Figure 7: Triton pseudo-code for Algorithm 1.

## E  Additional Experiments

We conducted an additional experiment to test whether this trend holds for larger models. We evaluated the Llama 13B model on the same downstream tasks and observed that the performance degradation becomes even more minimal. This leads to the tentative conclusion that CATS becomes more effective for larger models.

Additionally, we compared the performance of CATS with Wanda Sun et al. (2023), a method that prunes weights on a per-output basis based on the product of weight magnitudes and input activation norms. As Wanda achieves unstructured sparsity of weights, it is important to note that this approach may not lead to wall-clock time improvements in practice.

| Model | WG | PIQA | SciQ | QA | HS | BoolQ | Arc-E | Arc-C | Avg |
|---|---|---|---|---|---|---|---|---|---|
| Llama2-13B | 0.7230 | 0.7905 | 0.9460 | 0.3520 | 0.6006 | 0.8055 | 0.7946 | 0.4838 | 0.6870 |
| CATS 50% | **0.7111** | **0.7884** | **0.9480** | **0.3540** | **0.6057** | 0.7706 | **0.7870** | **0.4812** | **0.6805** |
| CATS 70% | 0.6946 | 0.7862 | 0.9310 | 0.3360 | 0.5987 | 0.7546 | 0.7546 | 0.4497 | 0.6616 |
| CATS 90% | 0.5596 | 0.6828 | 0.5820 | 0.2080 | 0.3960 | 0.6416 | 0.4726 | 0.3029 | 0.4807 |
| ReLUfication | 0.4933 | 0.5522 | 0.2660 | 0.1440 | 0.2648 | 0.4355 | 0.2757 | 0.2483 | 0.3350 |
| Wanda | 0.7079 | 0.7873 | 0.9450 | 0.3200 | 0.5710 | **0.8122** | 0.7605 | 0.4300 | 0.6667 |

Table 5: Zero-shot performance of 13B-based models, CATS-based models for varying levels of sparsity, ReLUfication, and Wanda 50% across different benchmark datasets. CATS 50% significantly outperforms other techniques, which suggests CATS scales well with model size.

