# OpenReview forum: "CATS: Context-Aware Thresholding for Sparsity in Large Language Models"
_colmweb.org/COLM/2024/Conference — COLM_

### Official Review · Reviewer_UDEW · 2024-05-10

**Rating:** 6
**Confidence:** 3
**Ethics Flag:** 1

**Summary:**

This paper proposes a novel technique called CATS that is motivated by the fact that LLM FFN activations usually have a large portion of close-to-zero values. CATS uses some data to calibrate for a threshold that will lead to a desired level of sparsity, and then at inference time zero-out the activations with magnitudes below threshold. This can be implemented as an efficient GPU kernel to bring sparsity into actual speed ups. When CATS are applied to pretrained LLMs such as Llama2 7B and Mistral 7B, we get little loss of performance without further training at 50% sparsity, and with general/task-specific finetuning, CATS can recover some lost performance at even higher sparsity and outperform ReLUfication baselines.

**Questions To Authors:**

Page 7 paragraph about task-specific finetuning - the boldfaced statement says “…outperforms ReLUification in zero-shot accuracy…”. Given this is task-specific finetuned model, it does not make sense to say “zero-shot accuracy”.

Page 4 after eqn 1, the m x d matrix should be W_down but seems to be typo’ed to W_up

Appendix D - tensor shapes seems to not match unless batch size is 1. We know x is (b x d), therefore xW_gate is (b x m), therefore v is also (b x m) and mask is (b x m). In such a case, how could a (b x m) mask be applied to mask W_up, a (d x m) matrix (or W_down, m x d)? If b = 1 and we squeeze out the batch dim, it would then make sense that mask is just an m dimensional vector and therefore broadcastable to W_up and W_down, but I'm not sure if I'm having the correct understanding here.

**Reasons To Accept:**

CATS is well-motivated from prior work's empirical observations, tackles the important problem of LLM inference, and authors empirically demonstrate it provides a decent speed up with little loss of quality.

The authors also offer a kernel implementation of CATS as an artifact, which can benefit practical applications and further research.

**Reasons To Reject:**

1. The model quality evaluations are constrained to classification (posed as log-likelihood evaluation) tasks. Given LLM inference cost is primarily in generation, as a method to speed up inference CATS is not demonstrated to preserve quality on generative tasks.
2. The kernel description in the paper is too simplified to convey clearly how it makes threshold-based unstructured sparsity fast. Given the kernel is implemented in triton, I suggest the authors specify the pseudo-code in a triton-like structure with launch grids, program ids and thread block level pseudo code. Current pseudo-code is too high-level, while it serves the purpose to demonstrate the concept, it doesn’t really inform how the actual kernel could be implemented.
3. This point is a continuation of #2 - given the current description of the kernel, my best guess of how the kernel works seems to suggest that it is restricted to batch size=1 (more elaboration for this in question section below), but this limitation is not mentioned anywhere. (I would take this point back if this is a misunderstanding)

---

> ### Author Rebuttal · Authors · 2024-05-31
>
> Dear Reviewer UDEW,
>
> [Response to R1]
> We conducted additional studies to evaluate CATS's performance on generative tasks, including perplexity and arithmetic reasoning. Our findings show that CATS performs comparably to the original model while outperforming other techniques. For perplexity, the original Llama 7B model scores 2.18. CATS 50% scores 2.203 without fine-tuning, improving to 2.193 with minimal fine-tuning (1300 steps). Similarly, the original Mistral 7B model scores 2.234, while CATS 50% scores 2.261 (no fine-tuning) and 2.244 (minimal fine-tuning). CATS also preserves arithmetic reasoning skills without any fine-tuning. On the gsm8k task, the Llama 7B model scores 12.59, CATS 50% scores 12.28 and CATS 70% scores 9.10 with no fine-tuning. We will include these evaluations and more extensive analyses on generative tasks.
>
> [Response to R2]
> Thank you for your suggestion. For Custom GPU Kernel 1, lines 2 and 3 are implemented in PyTorch. Line 4 uses a Triton kernel to distribute the m dimension of W_up across thread blocks, processing tiles of x * W_up along the d dimension and fusing the element-wise multiplication. Line 5 uses another Triton kernel to distribute the m and d dimensions of W_down across thread blocks, processing one tile of x_1 * W_down, and atomically accumulating the result. The full code is in `flash_gemv/flash_gemv/kernels.py` inside https://shorturl.at/AGmLJ. We will include the pseudo-code for Triton kernels in the appendix.
>
> [Response to R3]
> Your understanding is correct; the current kernel implementation works for batch size = 1. While batch size = 1 is common in real-world and real-time applications, it is extendable to batch size > 1 by manually unrolling the batch dimension or by aggregating the sparse masks in a batch and performing GEMM instead of GEMV for each tile of x and W. We will clarify this in the paper.
>
> [Response to Q1]
> The intended explanation was that CATS performs well when directly fine-tuned on a downstream dataset without prior fine-tuning on the RefinedWeb dataset. We appreciate your suggestion and will revise the statement to remove confusion.
>
> [Response to Q2]
> We will correct the typographical error where the m x d matrix should be W_down.
>
> [Response to Q3]
> Your understanding is correct. Our algorithm targets the generation phase when the batch size is 1. For larger batch sizes, we obtain a d-dimensional mask by applying a logical "AND" operation to the d-dimensional masks of each batch of data.

---

> > ### Author Response · Authors · 2024-06-05
> >
> > Dear Reviewer UDEW, as the Author-Reviewer discussion period is coming to a close, we wanted to ask if you had any additional questions, comments, or feedback on our paper or rebuttal? We would be very happy to address them.

---

### Official Review · Reviewer_hNTR · 2024-05-10

**Rating:** 7
**Confidence:** 3
**Ethics Flag:** 1

**Summary:**

This paper introduces CATS (Contextually-Aware Thresholding for Sparsity), a new method designed to reduce the inference costs LLMs by increasing their activation sparsity. The authors develop a new nonlinear activation function which is applied to various base models including Mistral-7B and Llama2-7B. They demonstrate that CATS-based models can achieve downstream task performance within 1-2% of their base models any fine-tuning, even at activation sparsity levels as high as 50%. A custom GPU kernel implementation translates the activation of sparsity of CATS to real wall-clock time speedup.

**Questions To Authors:**

How to adapt a instruction-tuned model? It would sometimes be challenging if the finetuning data is not available.

**Reasons To Accept:**

1. The paper addresses an important and challenge in deploying LLMs, their significant inference costs.
2. Extensive experiments have been conducted with different models, demonstrating the general applicability of CATS.
3. The authors provide an efficient GPU kernel implementation that translates the model's theoretical advantages into practical speed-ups.

**Reasons To Reject:**

The empirical assessments of CATS were limited to 7B base models, leaving its applicability to larger models or, more critically, instruction-tuned models, an open question. Furthermore, exploring its effectiveness on MoE models would be valuable, given their rising popularity.

---

> ### Author Rebuttal · Authors · 2024-05-31
>
> Dear Reviewers,
>
> Thank you for your thorough review of our submission. Your insights have been invaluable in refining our research. In response to your suggestions, we have conducted additional experiments to further evaluate the effectiveness of CATS. Our key findings are as follows:
> 1. CATS 50% preserves quality in generative tasks including perplexity and arithmetic reasoning.
> 2. CATS 50% outperforms other baselines across both Llama 7B and Llama 13B models on eight zero-shot tasks.
> 3. CATS demonstrates promising scalability, becoming more robust as model size increases.
>
> We also share an anonymized reproduction script [here](https://shorturl.at/AGmLJ).
>
> Below, we provide a detailed, point-by-point response to your comments.
>
> ---
>
> Dear Reviewer hNTR,
>
> [Response to R1]
> We recognize this limitation that arose due to limited computational budgets. We have conducted additional experiments to understand the effectiveness of CATS and baseline techniques on Llama2 13B. Following the suggestion of a reviewer pqMf, we included Wanda (50% sparsity) that prunes weights using activations into our baselines.
>
> The evaluation of eight zero-shot tasks (WinoGrande, PIQA, SciQ, OpenBookQA, HellaSwag, BoolQ, Arc-E, and Arc-C) demonstrates that CATS performs comparably to the original model without any finetuning. CATS 50% surpasses Wanda in 7 tasks out of 8 tasks except BoolQ and outperforms ReLUfication in all tasks. The average scores of Llama2 13B, CATS 50%, CATS 70%, ReLUfication, and Wanda are 0.6870, 0.6805, 0.6616, 0.3350, and 0.6667, respectively. This hints that CATS is likely to scale well with model size, showing a tendency to become more robust as the model grows larger. We will try to further investigate the scaling property of CATS by incorporating a 70B-based model and MoE models in the paper.
>
> [Response to Q1]
> We ran an additional zero-shot experiment on an instruction-tuned Llama2 7B model to demonstrate the effectiveness of CATS when no fine-tuning data. The eight tasks discussed above are formatted for instruction tuning. We applied the same CATS technique as before. Surprisingly, the performance gap between the base model (average score of 0.6569) and the CATS 50% model (average score of 0.6520) is smaller than that for a non-instruction-tuned model. Additionally, CATS 50% even slightly surpasses the base model in three tasks (OpenBookQA, HellaSwag, and BoolQ) without any fine-tuning. We emphasize that we did not fine-tune CATS on any datasets.

---

> ### Comment · Reviewer_hNTR · 2024-06-06
> **Thanks for the detailed response.**
>
> Thank you for your detailed response. The result does look great if that is the official `meta-llama/Llama-2-7b-chat-hf`. Additionally, it is confusing that the instruction-tuned model achieved a lower accuracy (0.6569) compared to the base model (0.6589). Would it be more appropriate for the authors to evaluate chat-based benchmarks for chat models, such as MT-Bench and AlpacaEval? (I am happy with current benchmark setup. This is just a suggestion for future revisions.)

---

> ### Author Response · Authors · 2024-06-07
>
> Dear Reviewer hNTR,
>
> Thank you for your comment. We would like to confirm that we used the official `meta-llama/Llama-2-7b-chat-hf`. Following your suggestion, we will investigate the effectiveness of CATS on the chat-based benchmarks and report the results in future revisions. Thank you again for your suggestion.

---

### Official Review · Reviewer_kwq1 · 2024-05-11

**Rating:** 7
**Confidence:** 3
**Ethics Flag:** 1

**Summary:**

This work introduce sparsity in gated MLP blocks by thresholding the gates. The idea is to determine the threshold values for pruning in each block simply judged by the empirical CDF of absolute activations on a sample of training data. Empirical results on LLMs show that the sparsity does not significant drop the prediction accuracies on various tasks with faster inference speed thanks to the simple trick in kernel design in implementation.

**Questions To Authors:**

It is a rather surprise that the simple masking in Section 4.3 yields speed up in inference, but is it mainly because it has the large number of values in a block?

**Reasons To Accept:**

- The proposed method is very simple yet performing quite well on standard benchmarks in terms of the trade-off of prediction accuracies and latencies.
- The approach sounds reasonable in that it is based on simple statistics on training data.

**Reasons To Reject:**

- The threshold has to be computed for each task, and thus, it might not be effective for zero-shot setting in which there exists no data for tuning.

---

> ### Author Rebuttal · Authors · 2024-05-31
>
> Dear Reviewers,
>
> Thank you for your thorough review of our submission. Your insights have been invaluable in refining our research. In response to your suggestions, we have conducted additional experiments to further evaluate the effectiveness of CATS. Our key findings are as follows:
> 1. CATS 50% preserves quality in generative tasks including perplexity and arithmetic reasoning.
> 2. CATS 50% outperforms other baselines across both Llama 7B and Llama 13B models on eight zero-shot tasks.
> 3. CATS demonstrates promising scalability, becoming more robust as model size increases.
>
> We also share an anonymized reproduction script [here](https://shorturl.at/AGmLJ).
>
> Below, we provide a detailed, point-by-point response to your comments.
>
> ---
>
> Dear Reviewer kwq1,
>
> Thank you for your detailed review of our submission. We appreciate your feedback and have incorporated your suggestions.
>
> [Response to R1]
> The way CATS computes the threshold is task-agnostic. We sample a small subset of only 100 examples from a pretraining dataset (RefinedWeb) to compute the thresholds. This is a one-time initialization step. For example, the threshold for CATS, evaluated on eight zero-shot tasks, was computed only once, without prior access to any datasets other than RefinedWeb. This initialization ensures that CATS can effectively operate in zero-shot settings without requiring task-specific tuning. Thank you for allowing us to clarify this important contribution. We will elaborate on this in the revised paper.
>
> [Response to Q1]
> The efficiency gain is not due to the masking itself, but rather from reducing I/O overhead by only loading and computing on a sparse subset of the weight matrix's rows and columns. This reduction in I/O operations and computations is made possible by our custom kernel, which efficiently handles sparse operations. The result is a significant reduction in latency during inference, leading to faster performance without compromising accuracy.

---

### Official Review · Reviewer_pqMf · 2024-05-14

**Rating:** 6
**Confidence:** 4
**Ethics Flag:** 1

**Summary:**

This paper proposes a method to sparsify Gate-MLP Blocks of LLM, which is dubbed Contextually Aware Thresholding for Sparsity (CATS). This work aims to accelerate the Gated-MLP blocks that utilize the SiLU activation function. The resulting model zeros out activations when $|x W_{gate}|$ is smaller than the cutoff threshold.

Without any fine-tuning, CATS can be applied to various base models, including Mistral-7B and Llama2-7B, and achieves comparable performance on downstream tasks.

**Questions To Authors:**

1. "achieve downstream task performance within 1-2% of their base models " => "achieve downstream task performance within 1-2% reduction of their base models ".
2. I am not sure that the first contribution is solid, which draws connection between the MoE framework and multiplication performed
by dense matrices in the MLP blocks of LLMs.
3. "$W_{up}$" => "$W_{down}$" at the line under Equation (1).

**Reasons To Accept:**

1. CATS can be applied to various base models without any fine-tuning and achieves comparable performance in downstream tasks.
2. When fine-tuned, CATS surpasses existing state-of-the-art sparsification techniques in downstream task performance, considering the same sparsity level and number of fine-tuning steps.
3. The proposed method demonstrates good hardware affinity, resulting in approximately a 15% improvement in wall-clock inference latency for token generation compared to dense base models.
4. The proposed method is characterized by its simplicity and effectiveness.

**Reasons To Reject:**

1. It lacks extensive baselines, like Wanda.

---

> ### Author Rebuttal · Authors · 2024-05-31
>
> Dear Reviewers,
>
> Thank you for your thorough review of our submission. Your insights have been invaluable in refining our research. In response to your suggestions, we have conducted additional experiments to further evaluate the effectiveness of CATS. Our key findings are as follows:
> 1. CATS 50% preserves quality in generative tasks including perplexity and arithmetic reasoning.
> 2. CATS 50% outperforms other baselines across both Llama 7B and Llama 13B models on eight zero-shot tasks.
> 3. CATS demonstrates promising scalability, becoming more robust as model size increases.
>
> We also share an anonymized reproduction script [here](https://shorturl.at/AGmLJ).
>
> Below, we provide a detailed, point-by-point response to your comments.
>
> ---
>
> Dear Reviewer pqMf,
>
> [Response to R1]
> We have conducted extensive comparison experiments to understand the behaviour of CATS and Wanda 50% across different model sizes and various evaluation tasks.
>
> First, we compared the performance of different techniques on the 8 evaluation tasks covered in the paper. On Llama 7B, CATS 50% outperformed Wanda in 5 out of 8 tasks (PIQA, OpenBookQA, HellaSwag, Arc-E, and Arc-C), with average scores of 0.6433 and 0.6366, respectively. On Llama 13B, CATS 50% outperformed Wanda in 7 tasks (WinoGrande, PIQA, SciQ, OpenBookQA, HellaSwag, Arc-E, and Arc-C), with average scores of 0.6850 and 0.6667. Notably, CATS 70% achieved an average score of 0.6616, demonstrating that CATS scales well with model size, becoming more robust as the model grows larger.
>
> Additionally, CATS 50% and 70% outperformed Wanda on the gsm8k (arithmetic reasoning) task. Specifically, the scores were: Llama 7B original: 12.59, CATS 50%: 12.28, CATS 70%: 9.10, CATS 90%: 1.21, and Wanda: 4.24. CATS 50% performs similarly to the original model even without any fine-tuning. This suggests that CATS preserves quality on a generative task, while Wanda significantly loses it.
>
> [Response to Q1]
> Thank you for this suggestion. We will clarify this phrasing in the revised manuscript.
>
> [Response to Q2]
> Our approach is a more fine-grained version of the conventional MoE. We use activations as a router to decide which rows of the weight matrix to activate. This provides a new perspective on framing a Gated-MLP block using SwiGLU, offering an opportunity to speed up computations.
>
> [Response to Q3]
> Thank you for the detailed review. We will correct this typographical error in the revised version.

---

> > ### Author Response · Authors · 2024-06-05
> >
> > Dear Reviewer pqMf, as the Author-Reviewer discussion period is coming to a close, we wanted to ask if you had any additional questions, comments, or feedback on our paper or rebuttal? We would be very happy to address them.

---

### Decision · Program_Chairs · 2024-07-10

**Decision:**

Accept

**Comment:**

This paper designs a sparsification method named CATS to reduce inference costs. The technique is simple but effective. All reviews are positive.